# Hierarchical Tactile Sensation Integration from Prosthetic Fingertips Enables Multi-Texture Surface Recognition [note 1]

**DOI:** 10.3390/s21134324

**Published:** 2021-06-24

**Authors:** Moaed A. Abd, Rudy Paul, Aparna Aravelli, Ou Bai, Leonel Lagos, Maohua Lin, Erik D. Engeberg

**Affiliations:** 1Ocean and Mechanical Engineering Department, Florida Atlantic University, Boca Raton, FL 33431, USA; mabd2015@fau.edu (M.A.A.); paulr2017@fau.edu (R.P.); mlin2014@fau.edu (M.L.); 2Applied Research Center, Florida International University, Miami, FL 33174, USA; aaravell@fiu.edu (A.A.); lagosl@fiu.edu (L.L.); 3Electrical and Computer Engineering, Florida International University, Miami, FL 33174, USA; obai@fiu.edu; 4The Center for Complex Systems & Brain Sciences, Florida Atlantic University, Boca Raton, FL 33431, USA

**Keywords:** liquid metal, tactile sensor, surface feature recognition, prosthetic hand, robotic hand

## Abstract

Multifunctional flexible tactile sensors could be useful to improve the control of prosthetic hands. To that end, highly stretchable liquid metal tactile sensors (LMS) were designed, manufactured via photolithography, and incorporated into the fingertips of a prosthetic hand. Three novel contributions were made with the LMS. First, individual fingertips were used to distinguish between different speeds of sliding contact with different surfaces. Second, differences in surface textures were reliably detected during sliding contact. Third, the capacity for hierarchical tactile sensor integration was demonstrated by using four LMS signals simultaneously to distinguish between ten complex multi-textured surfaces. Four different machine learning algorithms were compared for their successful classification capabilities: K-nearest neighbor (KNN), support vector machine (SVM), random forest (RF), and neural network (NN). The time-frequency features of the LMSs were extracted to train and test the machine learning algorithms. The NN generally performed the best at the speed and texture detection with a single finger and had a 99.2 ± 0.8% accuracy to distinguish between ten different multi-textured surfaces using four LMSs from four fingers simultaneously. The capability for hierarchical multi-finger tactile sensation integration could be useful to provide a higher level of intelligence for artificial hands.

## 1. Introduction

The sensation of touch for prosthetic hands is necessary to improve the upper limb amputee experience in everyday activities [1]. Commercially available prosthetic hands like the i-limb Ultra (Figure 1), which has six degrees of freedom (DOF), and the BeBionic prosthetic hand that has five powered DOFs, demonstrate the trend of increasing prosthesis dexterity [2], yet these state of the art prosthetic limbs lack tactile sensation capabilities when interacting with the environment and manipulating objects. The absence of sensory feedback can lead to a frustrating problem when grasped objects are crushed or dropped since the amputee is not directly aware of the prosthetic fingertip forces after the afferent neural pathway is severed [3,4]. Human hand control strategies depend heavily on touch sensations for object manipulation [5]; however, people with upper limb amputation are missing tactile sensations. Significant research has been done on tactile sensors for artificial hands [6], but there is still a need for advances in lightweight, low-cost, robust multimodal tactile sensors [7].

Tactile sensors have evolved from rigid components to now being completely flexible elastomers which have many applications in robotics [8,9,10,11,12]. Flexible pressure sensors are compatible with conventional microfabrication techniques which make them promising candidates for a tactile sensing solution in human-robot interfaces. Recently, research groups focused on stretchable tactile sensors for a soft prosthetic hand [13], while other groups have increased the capability of the tactile sensors for multimodal sensations [14]. Liquid metal (LM)—eutectic gallium–indium—can be encapsulated in silicone-based elastomers to create several key advantages over traditional sensors, including high conductivity, compliance, flexibility and stretchability [15]. These properties have been previously exploited for three-axis tactile force sensors [16] and shear force detection [17]. While sophisticated tactile sensors such as the BioTac have been previously used to distinguish between many different types of textures [18] and to detect sliding motion [19], this paper is the first to demonstrate such capabilities with a LMS for a prosthetic hand (Figure 1), to the best knowledge of the authors.

A manipulator with the ability to recognize the surface features of an object can lead to a higher level of autonomy or intelligence [20,21,22,23,24]. Furthermore, this information can be useful to provide haptic feedback for amputees who use prosthetic hands—to reconnect a previously severed sense of touch [25]. For the recognition of surface texture, it is very important to choose a suitable tactile sensor [26,27,28,29,30,31], that will be paired with machine learning algorithms to classify surface features [32,33,34,35,36].

There are three sources of novelty in this paper. First, we will use the LMS to discern the speed of sliding contact against different surfaces. Second, we will show that LMSs can be used to distinguish between different textures. And third, we will integrate tactile information from LMSs on four prosthetic hand fingertips simultaneously to distinguish between complex multi-textured surfaces, demonstrating a new form of hierarchical intelligence (Figure 1). Furthermore, the classification accuracy of four different machine learning algorithms to perform these three tasks will be compared. 

## 2. Materials and Methods

### 2.1. Liquid Metal Tactile Sensor Operational Principle

The electrical resistance of the LMS changes in response to externally applied forces. The conductive LM injected into the microfluidic channels (Figure 1) changes resistance when an external force impacts the microfluidic channel dimensions (Appendix A). The microfluidic channels increase in length and diminish in cross-sectional area when an external force is applied. The relation between the changes in resistance and changes in the microfluidic channel dimensions can be described by:(1)$$R=\frac{\rho \;L}{A}$$
where *R* is the resistance across the terminals of the LM conductor and ρ is the resistivity. Geometric dimensions *L* (length of the LM conductor) and *A* (cross-sectional area of the microfluidic channel) change as external forces deform the sensor, producing a measurable change in the electrical resistance of the LM conductor (Appendix A). 

### 2.2. Microfabrication of Liquid Metal Sensor Mold

The LMS mold was microfabricated via photolithography. SolidWorks (SolidWorks, Waltham, MA, USA) was used to design the LMS microchannel cross-section with 400 µm × 100 µm dimensions. The design was then printed onto a photomask to project the ultraviolet waves to the right dimensions. The silicon wafer was prepared first to create the 100 µm layer thickness of the mold. SU 8 50 was spin-coated (Ni-Lo 4 Vacuum Holder Digital Spin Coater, Ni-Lo Scientific, Ottawa, ON, Canada) at 1000 RPM for 30 s with an acceleration of 300 rad/s^2^. Soft bake was done at 65 °C for 10 min then at 95 °C for 30 min. An OAI 800 Mask Aligner (OAI, Milpitas, CA, USA) was used to perform the photolithography process. The photomask was placed on top of the SU 8 50 coated silicon wafer and the ultraviolet energy of 375 mJ/cm^2^ was used to transfer the microchannel patterns from the photomask to the coated silicon wafer (Figure 2a). After exposure, the coated silicon wafer was post-baked on a hotplate at 65 °C for 5 min and then 95 °C for 10 min. The sensor pattern was developed using SU 8 developer (Figure 2b).

### 2.3. Liquid Metal Sensor Manufacturing Process 

The LMS has two layers, the top and the bottom layer. To manufacture the top and the bottom layers, Dragon-Skin 30 (DS-30, Smooth-On, Macungie, PA, USA) material was used since the DS-30 material is flexible. A spin coating technique was used to manufacture and bond the top and the bottom layers. DS-30 was spread atop the surface of a blank wafer after it was centered, and the spin coater was run at a speed of 500 RPM with an acceleration of 100 rad⁄s^2^ for 60 s to create an even layer of DS-30 that was 250 µm thick (Figure 2c). This was next put into an oven at 60 °C for 10 min to cure. This process was repeated twice to obtain 500 µm thicknesses for both the top (Figure 2d) and the bottom layers. 

To bond the top and the bottom layers together, a 7 µm layer of DS-30 was obtained by operating the spin coater at 4000 RPM with an acceleration of 500 rad⁄s^2^ for 60 s. This layer was left to air dry at room temperature for 7 min so that it was firm but not yet cured. The patterned layer was lain atop the curing layer to form the microfluidic channel cavity (Figure 2e). This was then put into the oven to cure for five minutes.

### 2.4. Liquid Metal Injection Process

Next, LM was injected into the sealed microfluidic channel (Figure 2f) using two 31-gauge syringes. The first syringe was used to inject the LM into the microchannel while the second syringe was used simultaneously to extract the air trapped within the microchannel cavity (Figure 3a). Solid core electrical wires (gauge 30) were stripped a length of 0.75 cm and inserted to establish the electrical connectivity with the liquid metal material and to complete the fabrication of the LMS (Figure 3i). 

### 2.5. Design and Assembly of Fingertip Tactile Sensor

Distal phalanx support structures were designed to integrate the LMS in the fingertips of the i-limb Ultra prosthetic hand. The support structures were designed using SolidWorks and 3D printed with ABS using Ultimaker S3 (Ultimaker, Waltham, MA USA), (Figure 3c–e). Each LMS was tied against the flat face of the distal phalanx support structure using nylon string. Afterward, vertical stabilizers were printed to hold the LMS support structures motionless as the DS-30 was poured into the two-part outer cast (Figure 3f–g). The two halves were aligned, and the mold was taped shut to cure for 24 h. After curing, the mold was opened, the LMS was detached from the mold, and flash was trimmed (Figure 3h).

### 2.6. Robotic System Configuration

The original, sensorless fingertips of the i-limb were removed and the new fingertips with LMSs (Figure 3h) were mounted onto the i-limb using the same connection points (Figure 4). The prosthetic hand with the LMSs were next mounted onto a six DOF robotic arm (UR-10 (Universal Robots, Odense, Denmark)) to slide the hand along different surfaces and textures at different speeds (Figure 1). The hand was attached to the arm via a 3D printed mechanical adaptor with a coaxial electrical connector and a prosthetic hand lamination collar (Ossur, Reykjavik, Iceland) for stability and electrical connectivity. The robotic arm was programmed using the teach pendant to repeatedly perform the sliding contact against different surfaces and textures. 

LM has high conductivity so the resistance of each LMS was approximately 1 Ω. A five-channel printed circuit board was designed to amplify the LMS signals using Wheatstone bridges (Appendix A). The amplification board was powered by the Teensy 3.6 microcontroller (PJRC, Portland, OR, USA) with 3.3 V. The LMSs were connected to the amplifier board and then sampled through the Teensy as a ROS node. The ROS master was initiated in Simulink (The Mathworks, Natick, MA, USA) and the Teensy 3.6 was a slave node that sampled and published the LMS signals to the master in Simulink. Data were recorded in Simulink with a 1 kHz sample rate (Figure 4).

### 2.7. Liquid Metal Sensor Calibration 

The LMS then was calibrated with an ESP-35 load cell (Transducer Technique, Temecula, CA, USA), (Figure 5a–c), Appendix A). In the calibration process, the UR-10 arm was used to press the i-limb prosthetic hand fingertips against the load cell to produce the calibration forces consistently for each fingertip. The load cell and the LMS were connected to the same ROS network and had their data streamed into Simulink. The Curve Fitting MATLAB application was used to create calibration equations to correlate the load cell data to the LMS data for each individual fingertip (Figure 5d and Appendix A).

### 2.8. Experiment Design

To create a well-controlled experiment, we designed four different textures that had one variable parameter: the distance between the ridges (d). Texture 1 (T1) has d = 1 mm, texture 2 (T2) has d = 2 mm, texture 3 (T3) has d = 3 mm, and texture 4 (T4) has d = 4 mm (Figure 6). Different combinations of these four textures were combined to form surfaces upon which the four LMSs on the four prosthetic fingertips slid. Each surface was designed using SolidWorks and 3D printed using Ultimaker 5 with PLA material. To establish a stable contact with the LMSs, each texture was designed with different heights on the surface, depending on which texture the different fingers were intended to contact, due to the mechanical form factor of the prosthetic hand.

#### 2.8.1. Individual Fingertip Sensors to Detect Texture and Speed of Sliding Contact 

The first goal of this paper was to use individual LMS signals to distinguish between different sliding speeds (20 mm/s, 60 mm/s, and 100 mm/s). The second goal of this experiment was to use individual LMS signals to recognize four different textures (T1, T2, T3, T4, Figure 6). To these ends, four different surfaces were 3D printed: S_T1_, S_T2_, S_T3_, and S_T4_. Surface S_T1_ had four copies of texture T1 for each fingertip to simultaneously contact. Likewise, surfaces S_T2_, S_T3_, and S_T4_ each had four copies of textures T2, T3, and T4 to respectively contact and slide across (Appendix A). We hypothesized that the slip speeds and inter-ridge spacing (d, Figure 6) of the different textures would impact the power spectral distribution of the LMS signals [3,4,17]. The mechanism to detect the textures and speeds was to use the spectral components of the LMS signals to train machine learning algorithms to distinguish between the time-frequency signatures specific to each texture and speed [4]. 

To achieve the first and the second goals, 20 sliding trials were collected for each of the four aforementioned surfaces (S_T1_–S_T4_) with each of the three sliding speeds (20 mm/s, 60 mm/s, and 100 mm/s), producing 960 datasets for classification. The robotic arm was programmed to press the LMSs on the four prosthetic fingertips onto each surface. A momentary delay separated the initial contact and the subsequent sliding motion to ensure that the machine learning algorithms were trained using only the portions of time containing the slip events (see also Appendix A). The data from each finger were used to train and test four different machine learning algorithms (KNN, SVM, RF, and NN) for two different purposes: speed and texture detection. For specific textures, the algorithms were trained to detect the three speeds. For a given speed, the algorithms were trained to detect the four different textures. 

#### 2.8.2. Hierarchical Touch Sensation Integration to Detect Complex Multi-Textured Surfaces 

The third goal of this paper was to simultaneously use the LMS signals from the four fingertips of the prosthetic hand to recognize complex surfaces comprised of multiple textures. The hierarchical approach relied first upon successful texture detection localized to individual fingertips. This textural information from the four individual fingertips was integrated together to produce a higher state of knowledge at the level of the whole hand regarding the spatial layout of the multi-textured surfaces. To achieve this third goal, ten different multi-textured surfaces (S_1_–S_10_) were 3D printed using permutations of the four textures randomly generated by the MATLAB randperm function (Appendix A). An example of a complex surface comprised of four different textures is shown in Figure 7a–d. For each of the ten surfaces, 20 trials were collected to test the ability of the machine learning algorithms to distinguish between the ten different complex surfaces comprised of randomly generated permutations of four different textures. A MATLAB program was written to use the detected texture at each of the four fingertips to predict the spatial layout of the multi-textured surface that was contacted. This prediction was compared to the known database of ten surfaces comprised of multiple textures. The percentage of correct predictions was used to establish a success rate metric to quantify the capability to distinguish between the multi-textured surfaces. The speed of slip for these experiments was 20 mm/s. 

### 2.9. Machine Learning Classification Approach 

LMS time-domain data (Figure 7e–g,k–n) were trimmed and labeled. Time-frequency features (spectrogram) were extracted from the time domain to train the machine learning classifiers in MATLAB (Figure 7h–j,o–r). The frequency features extracted from the time domain data were calculated using a 512-point FFT with 0.08 s frame length and a Hanning window with 90% overlap. After running the spectrogram analysis, the time-frequency power distribution matrix was used to train the four machine learning algorithms. 

The SVM algorithm created a hyperplane to separate the extracted features into different classes, in this case, surfaces upon which the LMS contacted [37]. The KNN calculated the shortest distance between a query and all the points in the extracted features and selected the specified k number closest to the query to vote for the most frequent class label [38]. The RF algorithm was designed with 500 trees to perform regression and classification tasks [39]. Its ensembles of tree structure classifiers were trained separately to create a forest with a group of decision trees. In general, the more trees in the forest the more robust the prediction which leads to higher reported classification accuracy. The NN Pattern Recognition toolbox in MATLAB was used to create and train the NN, and evaluate its performance using cross-entropy and confusion matrices [40]. A two-layer feed-forward network with 100 sigmoid hidden and softmax output neurons was used to classify the collected data. The network was trained with scaled conjugate gradient backpropagation.

For the NN, the collected data were subdivided into 3 categories: 70% training dataset, 15% validation dataset, and 15% testing dataset. The training dataset was used to train the network, and the network was tuned according to its error. The validation dataset was used to measure network generalization, and to stop training when generalization stopped improving, while the testing data set provided an independent measure of network performance after training. Training automatically stopped when generalization error stopped improving, as indicated by an increase in the cross-entropy error of the validation samples. 

For the KNN, SVM, and RF algorithms, the extracted time-frequency power distribution features data for slip were divided randomly into two parts, the training part which comprised 80% of the data, and the testing data which comprised the remaining 20%. The performance of any classification model will decrease significantly if the model is overfit, so the feature data were toggled randomly before training and testing the classifier models to prevent any overfitting. Each of the classifier models was run 10 times with randomized selection of the training-testing features and the average classification accuracy was reported.

## 3. Results

### 3.1. Liquid Metal Sensors Sliding Across Different Textures

Sample data from the LMS on the little finger sliding across texture T3 with three different speeds showed different characteristics in the time domain (Figure 7e–g). The spectrograms of these experiments showed a higher concentration of power in higher frequencies as the speed of sliding contact increased (Figure 7h–j). Illustrative data from the LMS on the middle finger sliding across the four different textures (T1–T4) with a speed of 20 mm/s showed noticeable differences in the sensor responses (Figure 7k–n). The spectrogram showed that decreasing ridge spacing (d, Figure 6) of the textures resulted in increasing power in high frequency components of the LMS signal (Figure 7o–r). These characteristic signatures related to speed of sliding contact and texture features were exploited by the machine learning classification algorithms.

### 3.2. Detected Speed of Sliding Contact with Individual Fingertip Sensors

The machine learning algorithms were able to distinguish between all the speeds with each finger with high accuracy (Figure 8b). The overall means and standard deviations of speed classification accuracies from all four fingers are shown in Table 1, Figure 8a. The SVM, RF, and NN all had nearly perfect accuracy > 99% to detect the different speeds of sliding contact on all four textures. A two-factor ANOVA indicated that the classifier accuracy for the KNN was significantly different from the other algorithms (*p* < 0.01). There were no significant differences among the RF, SVM, and NN (*p* > 0.05). The classification accuracy to detect the speed of sliding contact was significantly impacted by the textures (*p* < 0.01), likely because the KNN, SVM, and RF generally had lower accuracies with textures T1 and T3 than with textures T2 and T4 (Figure 8b). However, there was no significant interaction between the different classification algorithms and the textures (*p* > 0.05). 

### 3.3. Distinguishing between Different Textures with Individual Fingertip Sensors

The LMSs were tested to distinguish between the four different textures. The mean texture classification accuracy for each classifier is shown in Table 2, and the classifier with the highest overall accuracy was the NN with 97.8 ± 1.0% (Figure 9a). The NN had the highest overall accuracy at the 20 mm/s and 60 mm/s speeds; however, the RF algorithm outperformed the NN slightly at the 100 mm/s slip speed setting with classification accuracy of 98.3 ± 1.7%. A two factor ANOVA showed that the classification accuracy for all algorithms were significantly different from one another (*p* < 0.01). This is likely because there was a clear trend of increasing accuracy with the KNN, SVM, RF, and NN (Table 2) that was consistent with each finger (Figure 9b, Appendix A). However, there was no statistically significant impact upon the texture detection accuracy by the speed of sliding contact (*p* > 0.05). There was also no significant interaction between the classification algorithms and the sliding speeds (*p* > 0.05).

### 3.4. Hierarchical Tactile Sensation Integration to Distinguish between Complex Multi-Textured Surfaces

Ten surfaces comprised of multiple textures (Appendix A) were accurately classified using the LMSs from four prosthetic fingertips simultaneously. Examples of five different multi-textured surfaces are shown in the top of Figure 10 and another is shown in Figure 7a–d. The algorithm with the highest overall classification accuracy was again the NN with 99.2 ± 0.7% (Figure 10, Table 3). The NN had the highest overall classification accuracy with all the surfaces except surfaces S7 and S8, where the RF algorithm outperformed the NN slightly. The two-factor ANOVA indicated that the accuracies for all classification algorithms were significantly different (*p* < 0.01) and the different surfaces also significantly impacted the classification accuracies (*p* < 0.01). Furthermore, there was a significant interaction between algorithms’ classification accuracies and the multi-textured surfaces (*p* < 0.01). This interaction was likely caused by the RF outperforming the NN with surfaces S7 and S8 even though the RF had lower overall classification accuracy across all ten surfaces than the NN. Another reason for the significant interaction is that the SVM had better classification accuracy than the RF with surface S2, but lower classification accuracy averaged across all surfaces than the RF.

## 4. Discussion

People exhibit the trait of hierarchical control, where tactile sensations from multiple fingers are integrated by the central nervous system to synergistically manage dexterous object manipulation tasks [41]. This capability has recently been demonstrated for dexterous robotic hands to autonomously stabilize grasped objects [42]. People can also detect the speed of sliding contact with their fingertips, and use this information to reliably control slip of grasped objects [43]. Furthermore, people can also discriminate between two different surface textures simultaneously using two different fingers [44]. To enable a more natural feeling prosthetic hand interface, recent work has shown that people can distinguish between several different surface textures using transcutaneous electrical nerve stimulation conveying haptic feedback from a neuromorphic sensor array [25].

Building upon these prior accomplishments, our paper has shown that the LMSs can be reliably used to distinguish between four different textures and three different speeds of sliding contact. Furthermore, we have demonstrated the capacity for hierarchical tactile sensation integration from four fingertips simultaneously to detect differences between complex multi-textured surfaces. This demonstrated the hierarchy of single finger (low-level) and whole hand (high-level) perception, specifically in the form of individual textures (low-level) and complex surfaces comprised of multiple textures (high-level). In other words, tactile information from all the individual fingertips provided the foundation for a higher hand-level of perception enabling the distinction between ten complex multi-textured surfaces that would not have been possible using purely local information from an individual fingertip. We believe that these tactile details could be useful in the future to afford a more realistic experience for prosthetic hand users through an advanced haptic display, which could help prevent prosthetic hand abandonment by enriching the amputee-prosthesis interface [45,46,47]. In the context of a haptic display, where sensations of touch are conveyed from a prosthetic hand to an amputee via actuators or electrotactile stimulation [25], there are many possible approaches for the mapping of tactile sensations from the robotic sensors to the human sensations on a different portion of the amputee’s residual limb. While it is possible that the user could learn to directly interpret the artificial sensations of touch, it is not necessarily the optimal approach. Fundamental differences between robotic sensors/actuators and human mechanoreceptors [7] suggest that an intermediary step of classification via artificial intelligence, such as the work in this paper, could ameliorate human perception of the haptic display. This could be accomplished by mapping the artificial tactile sensations in a manner more conducive to human perception [5,25,48].

Beyond this, the highly successful texture, speed and surface feature classification results were obtained without any explicit force feedback loop. Illustrative differences between LMS signal amplitudes of the little finger (Figure 7e–g) and the middle finger (Figure 7k–n) had no significant impact on the overall successful classification rates with each finger (Figure 8 and Figure 9 and Appendix A). Therefore, this approach for speed, texture, and multi-textured surface recognition does not require a force feedback loop, which could be quite beneficial in the highly unstructured tasks of daily life where amputees may seek to explore surface textures without the cognitive burden of accurate force control of each specific fingertip of a prosthetic hand. 

The comparison of the four different machine learning algorithms showed that the NN generally had the highest overall classification accuracy. However, when using a single finger to detect the speed of sliding contact, there were no significant differences between the SVM, RF, and NN algorithms. Furthermore, significant interactions were observed between the accuracies of each algorithm and the complex multi-textured surfaces when demonstrating hand-level intelligence via hierarchical tactile sensation integration. In some cases, the RF slightly outperformed the NN and the performance of any single algorithm was not perfectly uniform across all cases. These nuanced results illustrate the importance of considering many operational factors prior to choosing an algorithm for real-time control, including computational expense, the potential for increased classification accuracy, and the intended use of the sensor, which could be for prosthetic hands or more broadly to fully autonomous manipulators.

## 5. Conclusions

Stretchable tactile sensors using liquid metal were designed and manufactured for the fingertips of a prosthetic hand. Three novel contributions were made with these new tactile sensors. The LMSs were able to reliably distinguish between different speeds of sliding contact and different textures with individual fingers. Beyond this, we demonstrated the capability for hierarchical tactile sensation integration from four fingertips simultaneously to distinguish between ten complex multi-textured surfaces. The NN produced the highest classification accuracy of 99.2 ± 0.8% to classify the multi-textured surfaces. Due to the compliant, lightweight nature of the LMS and high classification accuracy, this paper has demonstrated the feasibility of their application to robotic hands. Hierarchical tactile sensation integration from multiple fingers is a trait exhibited by people, and could be a useful technique for a haptic display to improve prosthetic hand functionality or to augment the intelligence of autonomous manipulators.

## Figures and Tables

**Figure 1 sensors-21-04324-f001:**
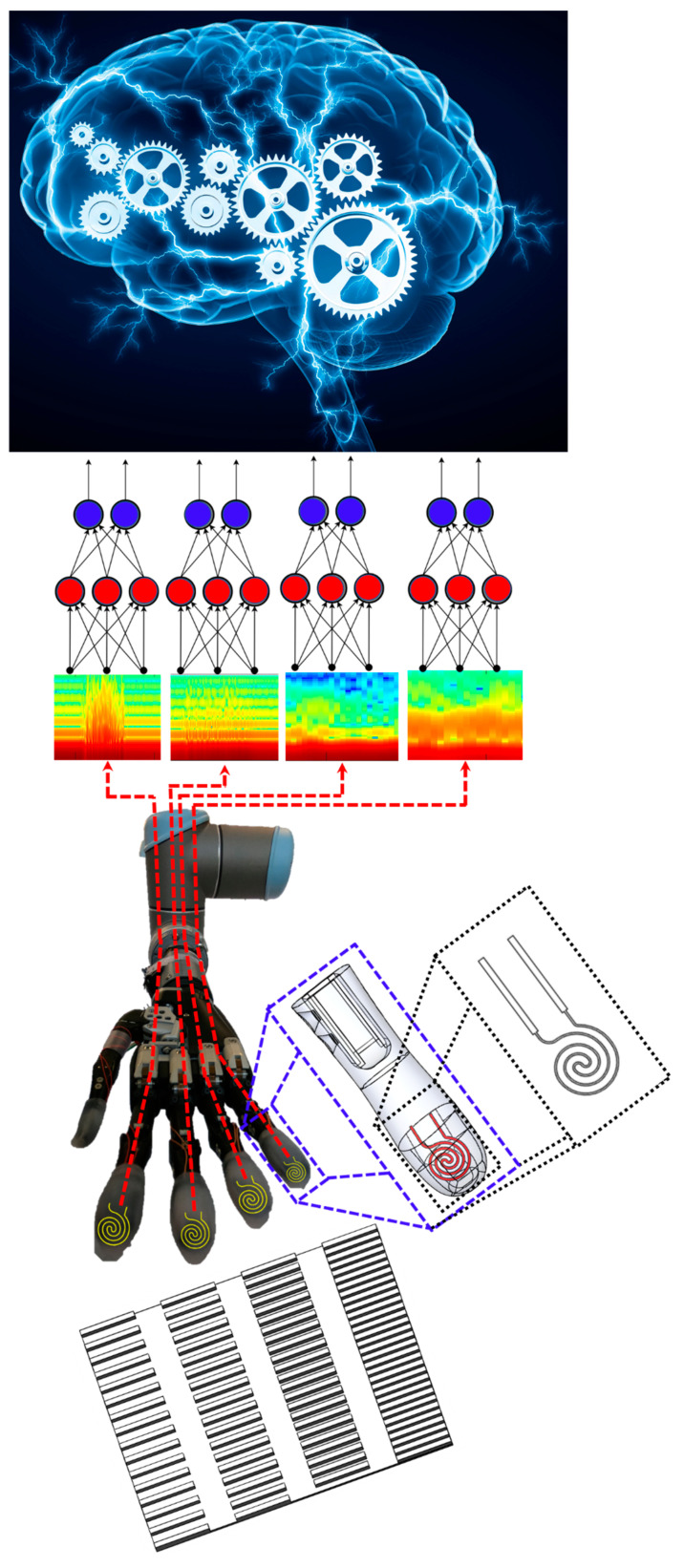
Liquid metal tactile sensors were integrated into the fingertips of the prosthetic hand. Individual LMS were used to distinguish between different textures and to discern the speed of sliding contact. Furthermore, LMS signals from four fingertips were simultaneously used to distinguish between complex surfaces comprised of multiple kinds of textures, demonstrating a new hierarchical form of intelligence.

**Figure 2 sensors-21-04324-f002:**
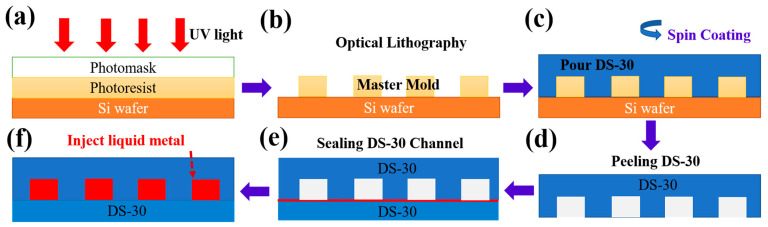
Liquid metal sensor manufacturing process. (**a**) Photolithography was used to manufacture the (**b**) Master mold. (**c**) Spin coating was used to manufacture the top and bottom layers. (**d**) The top part of the microfluidic channels was peeled off the mold. (**e**) A thin layer of DS-30 (red line) was used to bond and seal the top and bottom layers together. (**f**) After curing, liquid metal was injected into the sealed microchannels with a syringe. Adapted with permission from ref [4]. Copyright 2020 IEEE.

**Figure 3 sensors-21-04324-f003:**
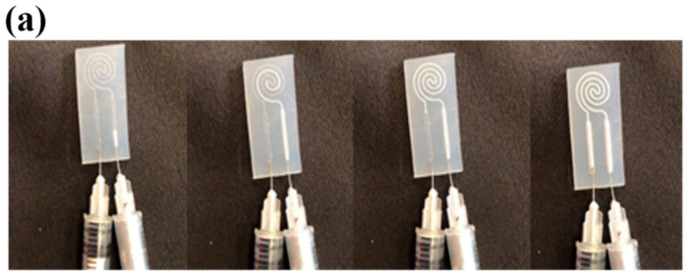
(**a**) The liquid metal was injected with one syringe while air within the microchannel cavity was simultaneously extracted with another syringe. (**b**) The LMS is highly stretchable (units of cm). (**c**) Fabrication of the first mold to create the inner part of the fingertip assembly: exploded view. (**d**) Assembled view for finger casting procedure. (**e**) Finger after removing from the cast. (**f**) 3D-printed finger-shaped cast. (**g**) 3D-printed inner finger part upon which the LMS was placed. (**h**) The completed fingertip with (**i**) integrated liquid metal sensor.

**Figure 4 sensors-21-04324-f004:**
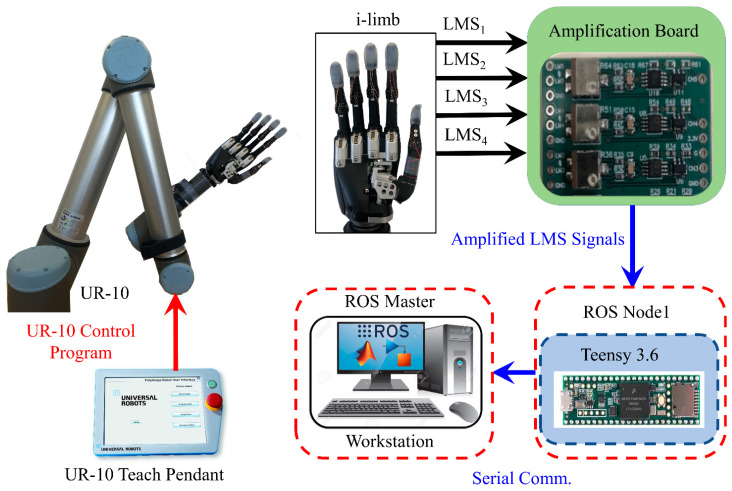
Robotic system configuration. The i-limb hand was attached to the robotic arm and the LMS tactile sensors were embedded in the fingertips. The LMS tactile sensor signals were amplified using the amplification board and recorded in MATLAB/Simulink via the ROS environment.

**Figure 5 sensors-21-04324-f005:**
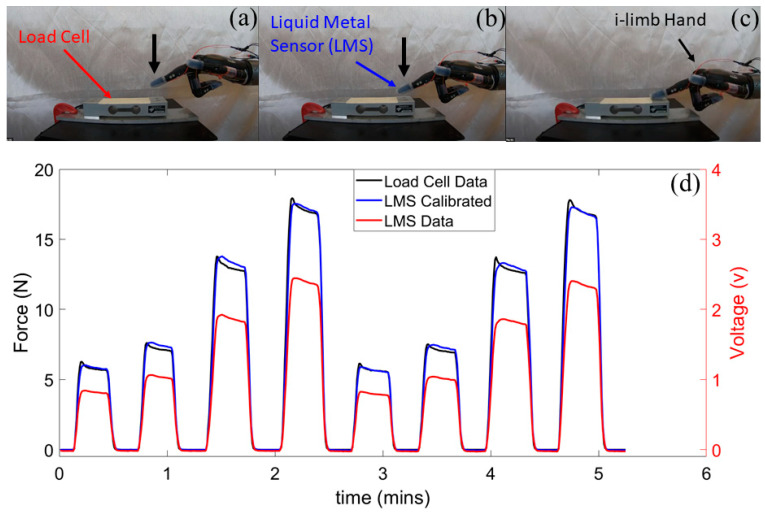
LMS calibration process. (**a**–**c**) The UR-10 robotic arm was used to press the LMS on the fingertip of the i-limb against a load cell as an external reference to (**d**) calibrate the LMS. See also Appendix A.

**Figure 6 sensors-21-04324-f006:**
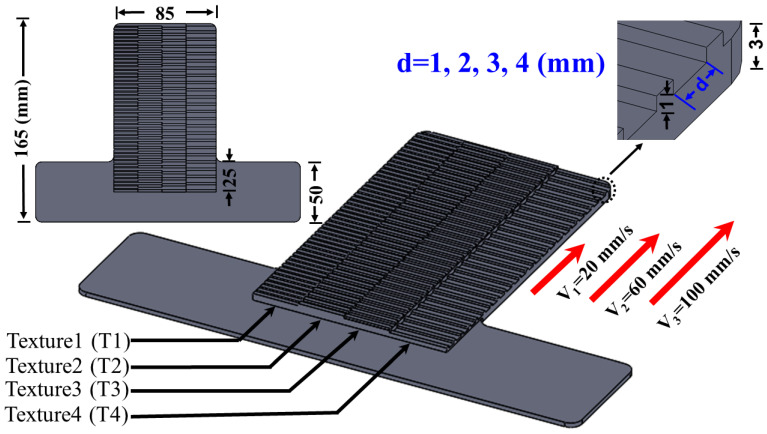
CAD model showing the four different texture dimensions and the three different sliding speeds. Units of mm.

**Figure 7 sensors-21-04324-f007:**
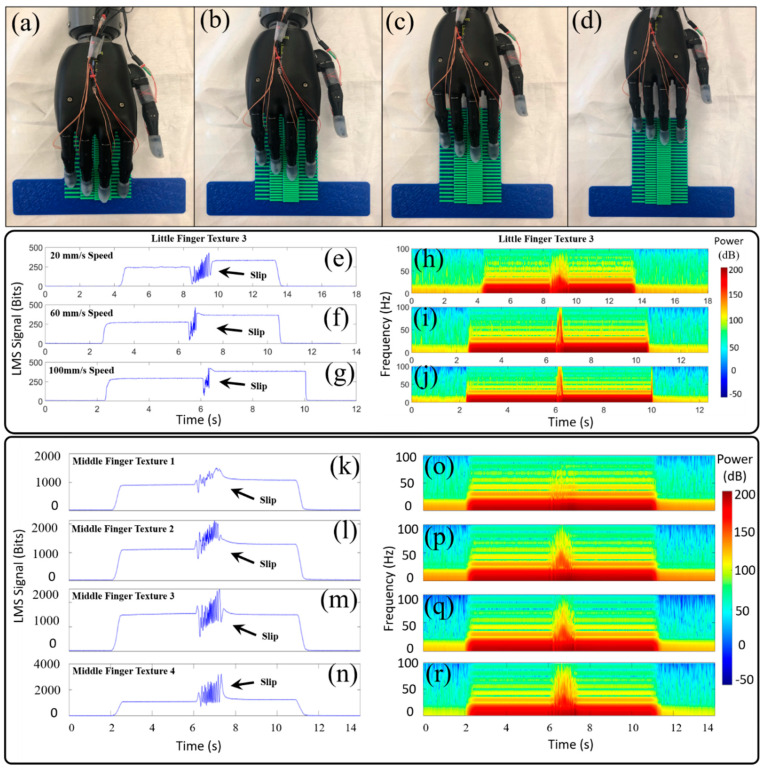
(**a**–**d**) The prosthetic hand with four LMSs slid while in contact with the multi-textured surface. (**e**) Illustrative data from the little finger LMS showed different responses when sliding on texture 3 at 20 mm/s, (**f**) 60 mm/s, and (**g**) 100 mm/s. (**h**) Corresponding spectrograms showed increasing power concentrations in higher frequency bands as the sliding speed increased from 20 mm/s to (**i**) 60 mm/s and (**j**) 100 mm/s. (**k**) Representative time domain LMS signals from the middle finger showed different activation patterns as it slid at 20 mm/s on texture 1 (**l**) texture 2, (**m**) texture 3, and (**n**) texture 4. (**o**) Corresponding spectrogram features revealed different frequency-domain signatures specific to texture 1, (**p**) texture 2, (**q**) texture 3, and (**r**) texture 4.

**Figure 8 sensors-21-04324-f008:**
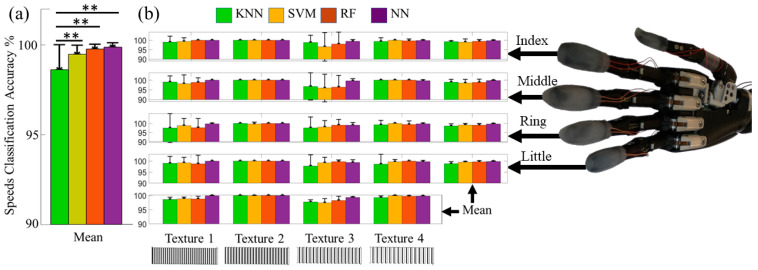
(**a**) The mean classification accuracy results from all four fingers to distinguish between different sliding speeds were > 99% for the SVM, RF, and NN algorithms. (**b**) Individual finger classification accuracies to detect the speed of sliding contact on specific textures were > 95% in all cases. ** *p*-value < 0.01.

**Figure 9 sensors-21-04324-f009:**
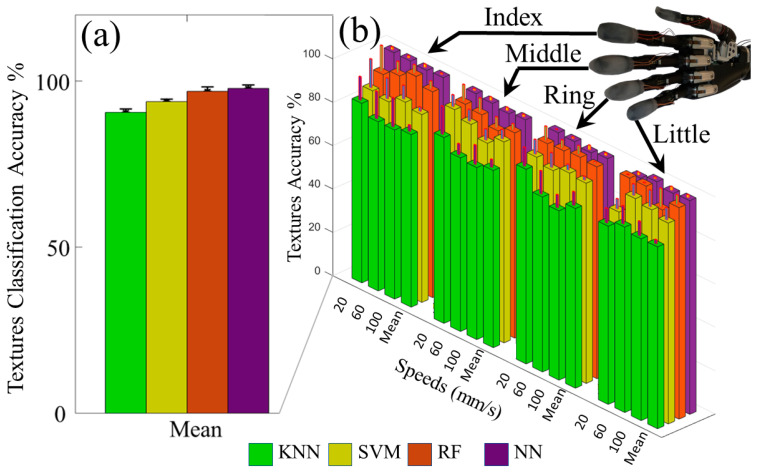
(**a**) Overall classification results to distinguish between different textures with three different speeds of sliding contact. (**b**) Classification accuracy for each finger to detect the correct texture with three different speeds of slip. (See also Appendix A).

**Figure 10 sensors-21-04324-f010:**
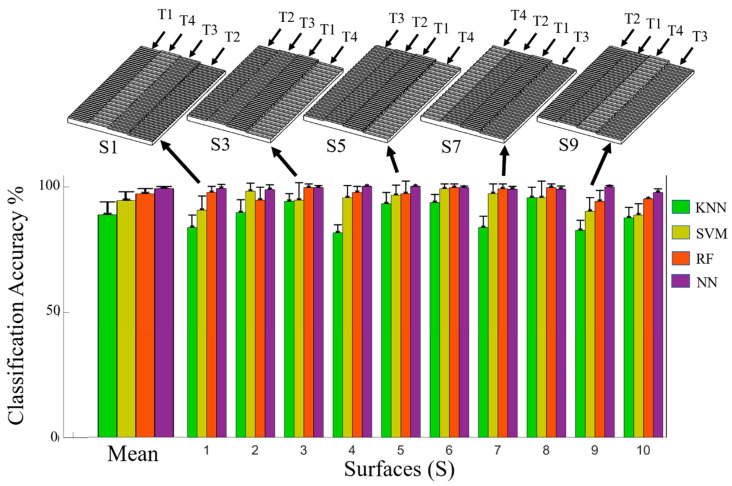
Classification results to detect 10 different complex multi-textured surfaces using four fingertip sensors simultaneously. Examples of five different multi-textured surfaces are shown on the top. On average, the NN had the highest classification accuracy for this new form of hierarchical tactile sensation integration.

**Table 1 sensors-21-04324-t001:** Mean classification accuracy for detected speed of sliding contact.

Classification Algorithms	Accuracy
K-Nearest Neighbors (KNN)	98.5 ± 1.3%
Support Vector Machine (SVM)	99.4 ± 0.4%
Random Forest (RF)	99.6 ± 0.2%
Neural Network (NN)	99.7 ± 0.2%

**Table 2 sensors-21-04324-t002:** Mean classification accuracy for detecting different textures with three different speeds using each finger individually.

Classification Algorithm	Accuracy
K-Nearest Neighbors (KNN)	90.5 ± 1.1%
Support Vector Machine (SVM)	93.8 ± 0.7%
Random Forest (RF)	96.8 ± 1.4%
Neural Network (NN)	97.8 ± 1.0%

**Table 3 sensors-21-04324-t003:** Mean classification accuracies to detect different multi-textured surfaces via hierarchical tactile sensation integration from four fingertip sensors simultaneously.

Classification Algorithms	Accuracy
K-Nearest Neighbors (KNN)	88.6 ± 5.1%
Support Vector Machine (SVM)	94.3 ± 3.4%
Random Forest (RF)	97.0 ± 2.1%
Neural Network (NN)	99.2 ± 0.8%

## Data Availability

The data presented in this paper will be made available upon reasonable request.

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
