# Peer review of "Hierarchical Tactile Sensation Integration from Prosthetic Fingertips Enables Multi-Texture Surface Recognition†"

_sensors, 2021, doi:10.3390/s21134324_

Round 1

Reviewer 1 Report

The authors report the synthesis of stretchable tactile sensors using liquid metal were designed and manufactured for the fingertips of a prosthetic hand and the resulting stretchable tactile sensors exhibits highest classification accuracy of 99% ± 0.76. This work is well carried out and written in good organization, thus interesting for prosthetic hands applications. The manuscript is suggested to be accepted for publication in Sensors.

Author Response

Thank you for your review of our manuscript.

Reviewer 2 Report

This manuscript demonstrates stretchable tactile sensors based on liquid metal for the fingertips of a prosthetic hand. The prosthetic hand with LMS sensor can successfully differentiate differences in surface textures and scanning speed. However, the authors need to add additional analytical results to support their claims. Specifically, I have the following comments.

  1. I would recommend adding figure that shows pressure-sensing mechanism of LMS sensor in Figure 1. In addition, I could not find the mechanism of texture detection in this manuscript.

  1. Why the size of texture and scanning speed affect to frequency in Figure 7?

  1. When the sensor contacts the material with texture, there are strong peaks in the frequency under ~30 Hz (Figure 2h-j and Figure 2o-r). From Figure 2e-g and Figure 2k-n, the sensor just contact the material and do not slide across the material. What is the reason?

  1. In the Figure 2e-g, the LMS signals are ~250. However, in the Figure 2k-, the sensor signals are ~1000. Because the pre-applied force will highly affect to the texture-sensing performances, the pre-applied force should be equally controlled for each case.

  1. Similar with #4, pre-applied force will be another factor that can control the texture-sensing sensitivity. I recommend adding additional data to show how the pre-applied force affects texture detection.

Reviewer 3 Report

The paper presents a tactile sensor based on liquid metal and its application to texture recognition.

The paper presents interesting results and ideas. Nevertheless, there are a few significant questions that should be clarified and eventually changed in the paper before being published, in my opinion:

1.- It is said that the system is able to distinguish between different speeds. Conversely, it is able to detect different textures at a given speed. Therefore, the dataset comprises data from the exploration of different textures at different velocities, and the classifiers are able to classify the specific texture that has been explored at a specific speed. Please confirm this and clarify this in the text, it is a bit confusing and could be interpreted as the system is able to provide the speed.

2.- In the paper (even in the title) it is highlighted the hierarchical tactile sensation integration. However, no explanation of the hierarchical approach can be found in the text. It is mentioned in sections 2.8.2 and 3.4, but the content of this quite brief sections does not describe in my opinion any hierarchical approach. The textures are explored with four fingers and the data from the four sensors are processed with the classifiers, but it is no clear if the processing is hierarchical or not.

3.- The approach of performing the FFT plus the use of classifiers is not new. Moreover, the textures are not complex (I mean those explored with each finger), it would have been good to see results from more complex textures being explored by an individual finger. Nevertheless, the proposal of the sensor, its fabrication and integration in the system is interesting, as well as the results in the application of texture detection.

Other comments:

1.- Some details of the electronics are welcome (signal conditioning circuit schematic and component data).

2.- section 2.7: I guess this fitting is performed for every finger. Please clarify this and if the curve is different for each sensor.

3.- In my opinion, it is clearer to say “perceptron” instead of NN (I think it is the same algorithm). Neural networks can be seen as a large set of algorithms.

4.- Section 3.2, page 11, lines 278-279. Please provide an interpretation of the results. This could be said for other sections.

5.- Section 3.3: Lines from 292 to 295 are confusing, it seems that they say that success rate and accuracy are not the same. Please clarify it.

Presentation:

1.- The style can be improved. Sometimes there are sharp transitions between sections that are closely related. For instance, from section 2.2 to section 2.3.

2.- Section 2.8.1 is not clear. I mean it is not understood clearly how are the textures ST1, ST2, ST3 and ST4 and why you need them (it seems that you used the data from the four fingers even in the ‘individual exploration’). Please clarify this.

3.- Please change the kind of graph to present the information of the results (Figure 8 and Figure 9). It is difficult to appreciate the differences in performance in a 3D graph such as that in Figure 8(b)

Reviewer 4 Report

Dear authors,

Thank you for submitting the manuscript Hierarchical Tactile Sensation Integration from Prosthetic Fingertips Enables Multi-Texture Surface Recognition. In the manuscript the authors present a new sensor designed for improving the tactile sensation of prosthetic hands. The authors test the sensors by attaching them to the fingers of a prosthetic hand and then slide the fingers over different surfaces. Different machine learning algorithms are capable of distinguishing the surfaces with very high accuracy.  

Comments

In the Introduction, I was missing a comparison with the state of the art. What is the hypothesised benefit with the LMS compared to other sensors?

Figure 1, the neural network-like architecture connects to the brain, but how do the authors envision that the output of the classification algorithm is purveyed to the user? Could the user not learn to distinguish the output of the sensors without using the classification algorithm?

Methods, calibration: Is the calibration done once, or does it need to be repeated at certain intervals?

Methods, surfaces: the different surfaces are all rough and quite distinct. What kind of real world surfaces do they resemble? Do they represent surfaces that prosthesis users would like to distinguish?

Methods, multi-textured surfaces: the “lanes” of the different surfaces are uniform. Did the authors investigate if they could distinguish changes in the surface? I would argue, that an important aspect is to detect surface changes (e.g. a change from the 1 mm surface to the 2 mm surface) as this would represent a possible real life situation. If the authors did not investigate this, then please elaborate if the authors think that the LMS would be capable of doing this.  

Results: the results show very high accuracies. Do the authors think a prosthesis user would be able to reach the same accuracies? Have the authors tried distinguishing surfaces with sub-millimetre spacing?

In the discussion the authors state: We believe that these tactile details could be useful in the future to afford a more realistic experience for prosthetic hand users through an advanced haptic display, which could help prevent prosthetic hand abandonment by enriching the amputee-prosthesis interface [45-47]. I am not sure what the authors mean by this statement in the light of the manuscript. How do the authors envision the use of the LMS by prosthesis users? Can the LMS provide “realistic” sensations? What is meant with the amputee-prosthesis interface? The socket? Or something else?

Round 2

Reviewer 3 Report

Thank you for addressing my comments.

Reviewer 4 Report

Dear authors, 

I am happy with the changes you made and will accept the manuscript in its current form.